# Human ACE2 peptide-mimics block SARS-CoV-2 pulmonary cells infection

Philippe Karoyan [1,2,3✉], Vincent Vieillard[4], Luis Gómez-Morales [1,2], Estelle Odile[1,2], Amélie Guihot[5,6], Charles-Edouard Luyt[7], Alexis Denis [8], Pascal Grondin [8] & Olivier Lequin[1]

In light of the recent accumulated knowledge on SARS-CoV-2 and its mode of human cells invasion, the binding of viral spike glycoprotein to human Angiotensin Converting Enzyme 2 (hACE2) receptor plays a central role in cell entry. We designed a series of peptides mimicking the *N*-terminal helix of hACE2 protein which contains most of the contacting residues at the binding site, exhibiting a high helical folding propensity in aqueous solution. Our best peptide-mimics are able to block SARS-CoV-2 human pulmonary cell infection with an inhibitory concentration ($IC_{50}$) in the nanomolar range upon binding to the virus spike protein with high affinity. These first-in-class blocking peptide mimics represent powerful tools that might be used in prophylactic and therapeutic approaches to fight the coronavirus disease 2019 (COVID-19).

[1] Sorbonne Université, École Normale Supérieure, PSL University, CNRS, Laboratoire des Biomolécules, LBM, 75005 Paris, France. [2] Sorbonne Université, École Normale Supérieure, PSL University, CNRS, Laboratoire des Biomolécules, LBM, Site OncoDesign, 25-27 Avenue du Québec, 91140 Villebon Sur Yvette, France. [3] χ-Pharma, 25 avenue du Québec, 91140 Villebon Sur Yvette, France. [4] Sorbonne Université, INSERM, CNRS, Centre d'Immunologie et des Maladies Infectieuses-Paris (CIMI-Paris), F-75013 Paris, France. [5] Assistance Publique-Hôpitaux de Paris (AP-HP), Hôpital Pitié-Salpêtrière, Département d'Immunologie, F-75013 Paris, France. [6] Sorbonne Université, Inserm U1135, Centre d'Immunologie et des Maladies Infectieuses, CIMI-Paris, F-75013 Paris, France. [7] Assistance Publique–Hôpitaux de Paris (AP-HP), Hôpital Pitié–Salpêtrière, Service de Médecine Intensive Réanimation, Institut de Cardiologie, F-75013 Paris, France. [8] Oncodesign, 25 Avenue du Québec, 91140 Villebon Sur Yvette, France. ✉email: philippe.karoyan@sorbonne-universite.fr

The coronavirus disease 2019 (COVID-19), caused by the severe acute respiratory syndrome-coronavirus 2 (SARS-CoV-2) has emerged as a pandemic, claiming at the time of writing more than 1.3 million deaths and over 57 millions confirmed cases world-wide between December 2019 and November 2020[1]. Since the SARS-CoV-2 discovery[2,3] and identification, the energy deployed by the scientific community has made it possible to generate an extraordinary wealth of information. However, to date, efficient therapeutics or drugs are lacking and prevention of the disease relies only on non-specific barrier measures[4]. Indeed, no specific drugs targeting the virus are available[5] yet, many clinical trials have been engaged with SARS-CoV-2 non-specific treatments[4]. The structural and biochemical basis of the mechanism of infection has been investigated, highlighting that the virus cell-surface spike protein of SARS-CoV-2 targets human receptors[6,7]. Human angiotensin converting enzyme 2 (hACE2) and the cellular transmembrane protease serine 2 (TMPRSS2) have been identified as major actors of the virus entry into human cells[8].

With the goal of preventing the SARS-CoV-2 from infecting human cells, blocking the interaction between hACE2 and the virus spike protein has been validated. Indeed, inhibition of SARS-CoV-2 infections in engineered human tissues using clinical-grade soluble ACE2 was recently demonstrated[9]. Likewise, an engineered stable mini-protein mimicking three helices of hACE2 to plug SARS-CoV-2 spikes[10] was described, but its capacity to block viral infection was not demonstrated. Although several in silico designed peptides were proposed to prevent formation of the fusion core[11,12], first attempts to design a peptide binder derived from hACE2 proved to be a difficult task, leading to mitigated results[13].

Thus, starting from the published crystal structure of SARS-CoV-2 spike receptor-binding domain (RBD) bound to hACE2[14], we designed peptide-mimics of the N-terminal hACE2 helix which interact with the spike protein. We report here the strategy implemented to optimize the design of our peptide mimics, their high helical folding propensity in water, their ability to block SARS-CoV-2 human pulmonary cell infection with an IC$_{50}$ in the nanomolar range and their binding to spike RBD with strong affinity. We also demonstrated the non-toxicity of our mimics at concentrations 150 times higher than the IC$_{50}$ on pulmonary cell lines.

## Results

**Design of peptides mimicking the helix H1 of hACE2.** We first examined the complex between hACE2 and the surface spike protein of SARS-CoV-2 (PDB 6m0j)[14] in order to highlight the important contacts and some relevant characteristics of the interacting hACE2 sequence (Fig. 1).

Twenty residues from hACE2 were identified[14] as to be in close contacts with the Spike protein, using a distance cut-off of 4 Å. These interactions occur mainly through the N-terminal α-helix **H1** of hACE2 (Fig. 1a). This α-helix (Fig. 1b), composed of 27 residues (from S19 to L45, Fig. 1c) contains 12 residues (highlighted in magenta in Fig. 1c) involved in hydrogen bonds, salt bridges, and van der Waals interactions (see Supplementary Table 1 for details)[14].

A Clustal multiple sequence alignment of viruses isolated in China, United States and France was performed (See Supplementary Fig. 1). From these analyses, we observed that all the randomly selected sequences were 100% identical at least in the ACE2 interacting interface. This highlights a highly conserved sequence for the portion of spike interacting with the α-helix H1 of hACE2, possibly because deleterious mutations at this interface would limit viral infectivity.

Our strategy was designing a peptide with a high helical folding propensity and retaining most of the binding affinity of hACE2 to the spike RBD of SARS-CoV-2, using natural amino acids[15]. Indeed, we preferred not to use complex chemical tools known to stabilize α-helix[16,17] in order to limit developability constraints. Our mimics were designed and optimized for binding, high helical content and low antigenicity, to avoid triggering a neutralizing immune response that would compromise the peptide therapeutic potential. A combination of the Agadir program[18,19], an algorithm developed to predict the helical content of peptides and the semi-empirical method reported by Kolaskar[20] to highlight the number of antigenic determinants, was iteratively used.

We observed that the N-terminal sequence of the **H1** helix, composed of four residues (S$_{19}$TIE$_{22}$), corresponds to a consensus N-capping box motif (SXXE)[21]. A capping box features reciprocal backbone-side-chain hydrogen-bonds favoring a helix initiation. Although this sequence does not adopt the H-bonded capping conformation in the crystal structure of the full protein, it could constitute a stabilizing element in the isolated helix when extracted from the protein context. These observations led us to

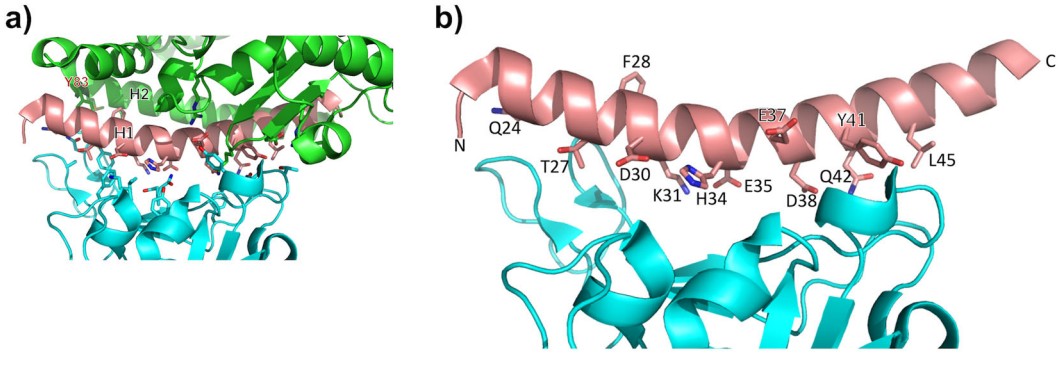

**c)** -S$_{19}$T$_{20}$I$_{21}$E$_{22}$E$_{23}$Q$_{24}$A$_{25}$K$_{26}$T$_{27}$F$_{28}$L$_{29}$D$_{30}$K$_{31}$F$_{32}$N$_{33}$H$_{34}$E$_{35}$A$_{36}$E$_{37}$D$_{38}$L$_{39}$F$_{40}$Y$_{41}$Q$_{42}$S$_{43}$S$_{44}$L$_{45}$-

**Fig. 1 Structure of the complex between hACE2 and the spike protein of SARS-CoV-2 (pdb 6m0j)[14].** **a** Contact residues of the hACE2 / SARS-CoV-2 spike interface. hACE2 protein is colored in green, apart from N-terminal helix H1 which is highlighted in salmon. SARS-CoV-2 spike protein is shown in cyan. **b** Residues of hACE2 H1 helix interacting with spike. **c** Sequence of hACE2 H1 helix showing the 12 interacting residues in magenta. Residue positions in green were considered as possible substitution sites for the helical peptide design.

keep 14 residues from the native **H1** helix of hACE2 as contact residues or putative stabilizing capping box. The 13 remaining residues that are not essential for the interaction were considered as possible sites for amino acid substitutions (Fig. 1c). We thus substituted non-essential positions by Ala and/or Leu residues which display higher helical folding propensities, and we calculated the peptide helical content after each substitution (Supplementary Table 2).

A peptide sequence optimization was then carried out to lower the antigenicity while keeping the helical propensity thanks to an iterative residue scanning and calculation of the helical content variation upon new substitutions (Supplementary Table 3). This strategy highlighted the influence of the residue N33 in the native sequence. Indeed, if the N33/L33 substitution systematically improved the helical content, it was always at the expense of antigenicity. Conversely, the L33/N33 substitution reduced the antigenicity at the expense of helical content. The solution was found by L33/M33 substitution which decreased the number of AD.

The **H1** helix of ACE2 adopts a kinked conformation in the crystal structure, leading to a distorted CO/HN hydrogen bond network between H34/D38 and E35/L39 residues. Therefore, we considered introducing a proline, as this residue is known to induce local kinks or distortions in natural helices[22,23]. D38 was classed as a contact residue, while L39 side chain is not involved in any direct interaction. Consequently, L39 position was selected for substitution by proline (peptide **P5**, Table 1).

In order to increase the helical content to a maximum level, this iterative study was also applied to longer peptide sequences starting from the 29-residue native one, albeit at the expense of antigenicity. Diverse combinations of *N*- and *C*-terminus capping groups were also considered (free extremities or *N*-acetyl, *C*-carboxamide groups).

Finally, we examined the possibility of promoting additional side chain contacts provided by ACE2 residues that do not belong to **H1** helix. Y83 residue in **H2** helix appeared as a good candidate as it lies very close in space to A25 in **H1** helix (Fig. 1a). Molecular modeling was carried out on **H1** helix analogs in which A25 was replaced by tyrosine or homotyrosine (*h*Tyr) residues (Supplementary Fig. 2). Calculations showed that *h*Tyr residue was able to project the phenol ring in an adequate orientation to mimic Y83 position. Of note, *h*Tyr is a natural amino acid[24].

Three peptides were selected as controls in our optimization process, **P1** (native sequence), **P1scr** (scrambled peptide from **P1**), and **Ppen** (described by Pentelute and colleagues in a longer biotinylated and pegylated construct and termed SBP1 as a putative spike binder[13]).

The results highlighting the progression in helical content and number of antigenic determinants are reported in Table 1 for the most relevant peptide mimics (see Supplementary Tables 2–4 for all the peptide-mimics that were designed and/or synthesized). These peptides were synthesized on a 5-mg to 20-mg scale from Fmoc-protected amino acids utilizing standard solid-phase peptide synthesis methods on rink amide resin (see Methods section).

**The designed peptides highlight an excellent correlation between calculated and experimentally determined helical content by circular dichroism in aqueous media.** The conformation of synthesized peptides in aqueous solution was investigated by circular dichroism (CD) spectroscopy[25]. Figure 2 shows the superimposed CD spectra of 12 peptides, including control ones, i.e., **P1** (native sequence), **P1scr** (Scramble), and **Ppen**. The CD spectra of peptides **P1** (native), **P1scr** (scrambled) are characteristic of a predominant random coil structure with a

negative minimum near 200 nm, as expected. Similarly, **Ppen** described as a helical peptide sequence[13] also adopted a random coil conformation in solution. For all other peptides, the CD spectra exhibit a canonical α-helix signature, featuring a double minimum around 208 nm and 222 nm, with the exception of the proline-containing **P5** peptide. The deconvolution of the CD spectra using DichroWeb[25] allowed us to estimate the helical population for each peptide, which is reported in Table 1. Overall, an excellent agreement was observed between the Agadir-computed values and the experimental helical population inferred from CD data. The native hACE2 **H1** helix sequence (peptides **P1**, **Ppen**) has a weak propensity to fold into an α-helix in aqueous solution (below 10%). In contrast, the sequence optimization led to **H1** analogs exhibiting a high helical propensity (between 50 and 80%). The introduction of a proline residue in peptide **P5** has a strong destabilizing effect on helical conformation (17%) whereas Leu/*h*Tyr substitution only led to a slight decrease of the helical content (**P7** versus **P6** and **P10** versus **P8**).

**Peptide mimics of hACE2 show high anti-infective efficacy and are devoid of cell toxicity.** To determine whether our peptide mimics of hACE2 **H1** helix block SARS-CoV-2 cell infection, antiviral assays[26] were performed (Fig. 3) with a SARS-CoV-2 clinical isolate obtained from bronchoalveolar lavage (BAL) of a symptomatic infected patient (#SARS-CoV-2/PSL2020) at Pitié-Salpêtrière hospital, Paris (France) (see Methods section). We first measured the inhibition of viral replication in Vero-E6 cell cultures exposed to 10 μM of the first set of peptide-mimics (**P2** to **P8**, **P1**, **P1scr**, and **Ppen** being used as controls), for 48 h (Fig. 3a). These preliminary assays helped us to identify two peptide-mimics that stand out (**P7** and **P8**) for their ability to block the viral infection, highlighting a potential role of *h*Tyr. This observation helped us in the peptide-mimics structure optimization process. Two new peptides were designed, **P9** and **P10** incorporating the *h*Tyr residue and evaluated with **P8** for their ability to block viral infection on Vero-E6 cells through the measurement of infectious virus production and viral genome (Fig. 3b and Supplementary Fig. 3)[27]. These peptides proved to be devoid of toxicity in Vero-E6 (Fig. 3c). In order to get insight on their ability to block viral infection on human pulmonary cells, Calu-3 cell line (ATCC HTB55) was chosen. This pulmonary epithelial cell line is commonly used as a respiratory models in preclinical applications[28] and SARS-CoV-2 has been shown to replicate efficiently in this cell line[29].

We first observed a dose-dependent reduction in virus titer (Fig. 3d) and then, using ELISA assays we evaluated the average median inhibitory concentration (IC$_{50}$) on Calu-3 cells for **P8**, **P9**, and **P10** to be 46 nM, 53 nM, and 42 nM respectively (Fig. 3e). Importantly, no cytotoxicity was observed in similarly treated uninfected culture cells at 10 μM, a concentration 150 times higher than the IC$_{50}$ (Fig. 3f). Collectively, these data demonstrate the high antiviral potency of peptide analogs **P8**, **P9**, and **P10**.

**The designed peptides bind to SARS-CoV-2 spike RBD with high affinity.** Finally, the peptides that were able to block cell infection with an IC$_{50}$ in the sub-μM range (**P8**, **P9**, and **P10**) were evaluated for their ability to bind to SARS-CoV-2 spike RBD (Fig. 4) using biolayer Interferometry (BLI) with an Octet RED96e system (FortéBio)[30]. hACE2 was used as a positive control (Fig. 4a).

Even though this technique presents some drawbacks[31] offering narrow signal windows with low molecular weight analytes[32] such as peptides, it remained useful to identify and

**Table 1 Sequences and properties of synthesized peptides.**

| Code | Sequence[a] | Predicted helical content%[b] | Predicted antigenicity (AD)[c] | Experimental helical content%[d] | % Inhibition of SARSCoV-2 replication at 10 µM[e] | SARSCoV-2 virus titer at 1 µM (PFU mL$^{-1}$) on Calu-3[f] | IC$_{50}$ (nM)[g] Calu-3 | $K_D$[h] (nM) |
|---|---|---|---|---|---|---|---|---|
| P1 | STIEE QAKTF LDKFN HEAED LFYQS SL-NH$_2$ | 6 | 0 | 7 | 9 | – | – | – |
| P1scr | AHLFS YLTTK EEQDN DAIFL QEFSK ES-NH$_2$ | 1 | 0 | 8 | 7 | – | – | – |
| Ppen | IEE QAKTF LDKFN HEAED LFYQS-NH$_2$ | 1 | 0 | 10 | 14 | – | – | – |
| P2 | SALEE QLKTF LDKFL HELED LLYQL SL-NH$_2$ | 64 | 1 | 70 | 16 | – | – | – |
| P3 | SALEE QLKTF LDKFL HELED LLYQL AL-NH$_2$ | 78 | 1 | 64 | 4 | – | – | – |
| P4 | Ac-SALEE QLKTF LDKFL HELED LLYQL AL-NH$_2$ | 66 | 1 | 70 | 24 | – | – | – |
| P5 | SALEE QLKTF LDKFL HELED PLYQL AL-NH$_2$ | 34 | 1 | 17 | 27 | – | – | – |
| P6 | SALEE QLKTF LDKFL HELED LLYQL ALAL-NH$_2$ | 88 | 1 | 80 | 6,5 | – | – | – |
| P7 | SALEE QYKTF LDKFL HELED LLYQL ALAL-NH$_2$ | n.a. | n.a. | 63 | 54 | – | – | – |
| P8 | SALEE QLKTF LDKFM HELED LLYQL AL-NH$_2$ | 68 | 0 | 70 | 91 | 0 | 46 | 24 ± 11 |
| P9 | SALEE QYKTF LDKFM HELED LLYQL SL-NH$_2$ | n.a. | n.a. | 53 | 93 | 0 | 53 | 0.09 ± 0.08 |
| P10 | SALEE QYKTF LDKFM HELED LLYQL AL-NH$_2$ | n.a. | n.a. | 56 | 95 | 0 | 42 | 0.03 ± 0.01 |

[a]The homotyrosine residue is depicted with the underlined letter Y in peptides P7, P9, and P10.
[b]Predicted helical content using Agadir program available at https://agadir.crg.es; n.a., not applicable (for hTyr containing sequences).
[c]Predicted antigenicity calculated at http://imed.med.ucm.es/Tools/antigenic.pl; n.a., not applicable (for hTyr containing sequences).
[d]Experimental helical content determined by circular dichroism (see Fig. 2).
[e]Mean percentage of inhibition on Vero-E6 cells of at least three independent experiments (see Fig. 3a and Supplementary Fig. 3).
[f]Mean viral titration on Calu-3 cells (see Fig. 3d (Fig. 3b for Vero-E6 cells)).
[g]IC$_{50}$ determined on Calu-3 cells only for peptides able to inhibit SARSCoV-2 cell infection.
[h]Dissociation constants with SARS-Cov-2 RBD-SD1 measured by BLI for peptides with IC$_{50}$ in the nM range. $K_D$ values are the mean of three independent determinations (P9 and P10).

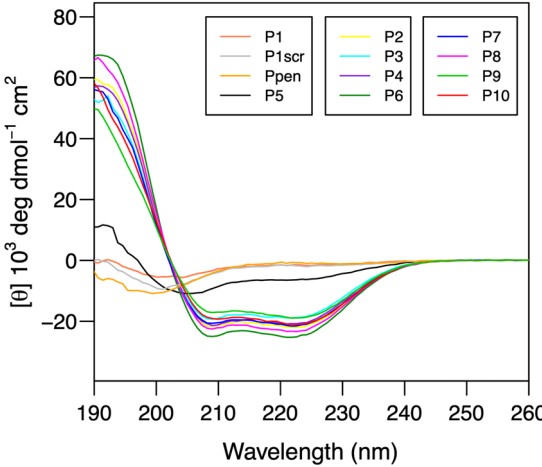

**Fig. 2 Far-UV CD spectra of synthesized peptides.** Samples of peptides **P1**-**P10**, **P1Scr**, and **Ppen** were prepared at a concentration of 60 µM in 50 mM sodium phosphate buffer at pH 7.4. CD measurements are reported as molar ellipticity per residue.

rank our binding mimics. In the conditions tested here, peptide **P1** does not bind to RBD when using 100 nM nor 10 µM peptide solutions (Fig. 4b). For peptides **P8**, **P9**, and **P10**, multiple concentrations experiments were performed and dose-dependent associations were observed (Fig. 4c–e). Of note, only association rates could be quantified accurately, the dissociation ones being very slow, highlighting strong binding properties for these mimics (Supplementary Table 6).

## Discussion
The current pandemic originated by SARS-CoV-2 causes an unprecedented health crisis. The medical world has found itself helpless in the face of this virus, having to deal with the absence of specific effective treatment. At the time of writing, preventive vaccines are not yet clinically approved, other antiviral drugs have been shown ineffective, and specific drugs addressing SARS-CoV-2 targets are lacking[4]. Among all possible viral targets, the virus spike protein/hACE2 interaction has been validated and the design of compounds able to block this interaction upon binding to spike protein is a promising approach. However, developing a specific drug at a pandemic speed is a hard task especially in the design of a small molecule. Indeed, beyond the time required for the identification and validation of a lead compound after a library screening, followed by structure-activity relationship studies and clinical development, small molecule drugs are associated with a high attrition rate partly due to their off-target toxicity observed during pharmacological studies.

Peptides appear here as a possible solution for design and development at pandemic speed. Peptides are widely recognized as promising therapeutic agents for the treatment of various diseases such as cancer, and metabolic, infectious, or cardiovascular diseases[33–35]. To date, ~70 peptide drugs have reached the market and 150 are currently under clinical development[33,36]. Special advantages that peptides show over small drugs include their high versatility, target-specificity, lower toxicity, and ability to act on a wide variety of targets[35] which are directly responsible for a greater success rate than small molecules (approval rate of around 20% versus 10%)[36,37]. The synthesis and the development of long therapeutic peptides (over 30 residues) are no longer a challenge, as highlighted by the success story of many GLP-1 analogs[38]. Their possible antigenicity can be evaluated using prediction tools in the design[20].

Of course, even for peptides, the development of a drug at a pandemic speed requires some considerations. Our aim was to design a peptide with reasonable helical folding propensity in water in order to mimic the **H1** helix of hACE2 in the protein context, considering that this helical folding is a prerequisite to compete with hACE2 upon interaction with viral spike protein. The design was realized using only natural amino acids[15] and avoided complex tools that are known and validated to stabilize α-helix[16]. Stabilizing α-helical structure of medium-sized peptide sequences (up to 15 residues) using only natural amino acids is a hard but achievable task[39]. Our choice was guided by the desire to build a simple peptide that can be upscaled quickly and easily, without technical constraints that may require laborious development. We also assumed that the use of mostly natural amino acids can facilitate the essential stages of the development of therapeutic tools in the event of success, particularly concerning pharmacokinetics, preclinical, and clinical toxicity aspects. We thought it would be a fair compromise between designing α-helix peptides with optimized binding affinity and developing an effective tool within short deadlines while integrating the constraints of developability to move a prophylactic device and/or a therapeutic peptide drug quickly in the clinic.

Using a combination of validated methods, we improved the helical folding propensity of the native α-helix extracted from the protein context. Thanks to leucine and alanine scanning (Table 1 and Supplementary Tables 2 and 3) a first set of peptides (**P1** to **P8**) were designed and synthesized. These peptides demonstrated to have a high helical content (up to 80% for **P6**). However, increasing this helical content to a maximum level led to increasing mean hydrophobicity and a hydrophobic moment (Supplementary Table 4) that proved to be detrimental to solubility and efficacy. The substitution of a leucine residue by the homotyrosine residue led to the peptide analog **P7** with a slight increase in solubility and a weak efficiency to block SARS-CoV-2 cell infection. In this first generation of peptides, the 27-residue **P8** peptide appeared to be highly soluble with a high helical folding propensity (70%), and an ability to block SARS-CoV-2 cell infection at 10 µM on Vero-E6 cells (Fig. 3a).

In order to improve the potency of our peptides, we designed a second set of mimics combining the properties of peptides **P7** and **P8**, i.e., **P9** and **P10**. If the Leu/$h$Tyr substitution led to a slight decrease in helicity, this was at the advantage of the mean hydrophobicity (Supplementary Table 4) also highlighted by lower HPLC retention times (Supplementary Table 5 and Supplementary Data 1). These peptides proved to be highly efficient in reducing SARS-CoV-2 viral titers (100% efficacy at 1 µM) on pulmonary cells with an $IC_{50}$ in the nanomolar range. This blocking property is related to their ability to bind to SARS-CoV-2 spike RBD with affinity estimated in the sub-nanomolar range (Table 1 and Supplementary Table 6). Finally, these peptides proved to be devoid of cell toxicity at 150 times the $IC_{50}$ concentration (Fig. 3c, f) highlighting their therapeutic potential.

In conclusion, we demonstrated here the feasibility of designing hACE2 peptide-mimics with high helical folding propensity in water. The folding propensity promotes interaction with spike RBD and blocks SARS-CoV-2 pulmonary cell infection. Devoid of cell toxicity even at high doses, these mimics could be considered for prophylactic or therapeutic purposes upon adequate formulation. Targeting prophylaxis first might shorten the drug development time scale. Delivered through a medical device such as a nasal or oral spray or as a sublingual tablet, these peptides could be aimed at blocking the infectivity of the virus in a preventive manner. Their biodistribution would be limited to the upper airways (nasal and oral cavity) before they are degraded in the digestive tracks without toxic residues. Their therapeutic use might also be considered formulated in that case as an inhaler to reach pulmonary cells.

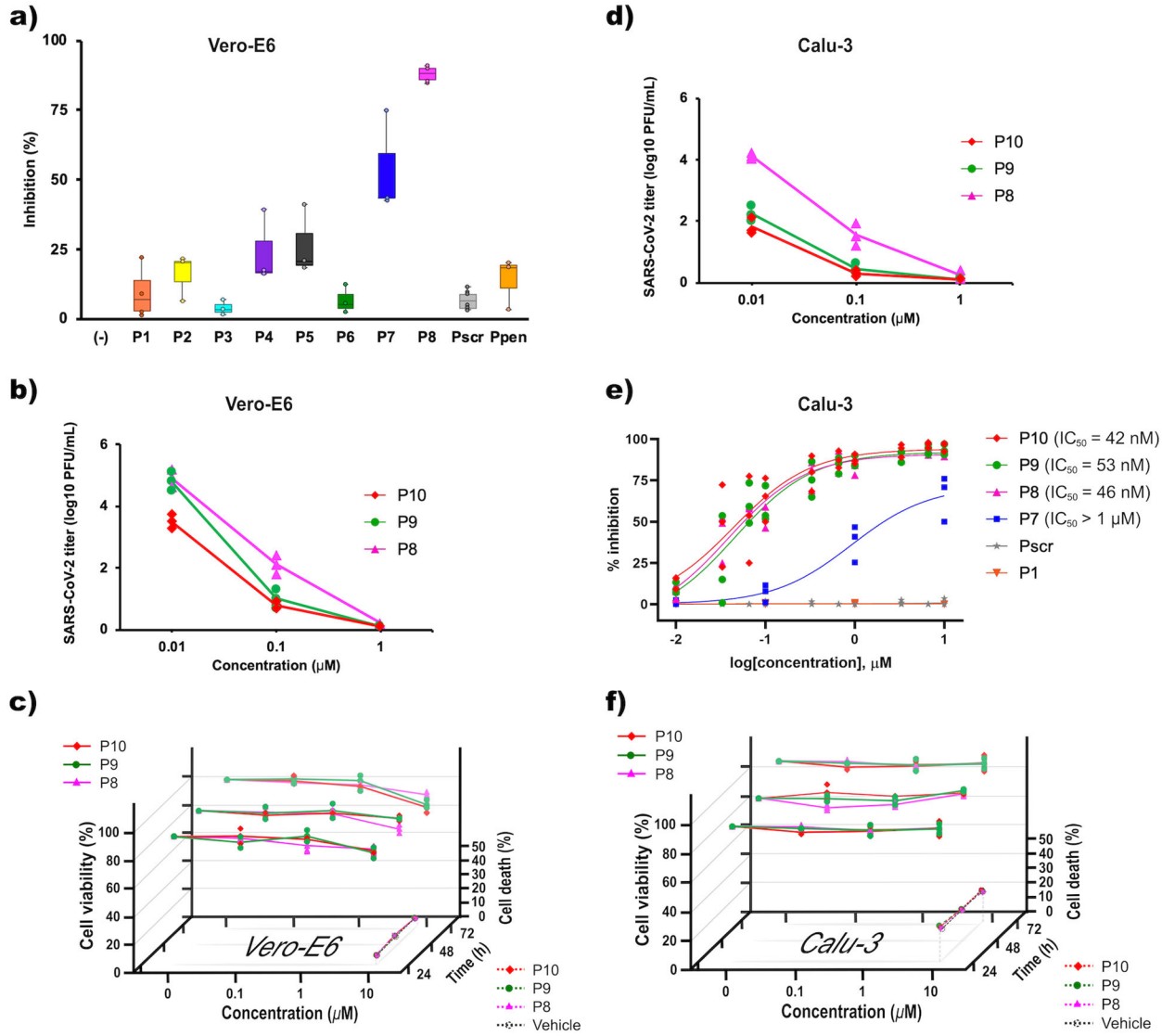

**Fig. 3 Peptide-mimics of hACE2 show high anti-infective efficacy and are devoid of cell toxicity. a** Percent inhibition of SARS-Cov-2 replication. Vero-E6 cells were infected with SARS-CoV-2/PSL2020 P#2 stock at a multiplicity of infection (MOI) of 0.1 in the presence of 10 peptides (**P1 to P8, P1scr**, and **Ppen** as controls) at 10 μM for 2 h. Then, the virus was removed, and cultures were washed, incubated for 48 h, before supernatant was collected to measure virus replication by ELISA. Data are combined from 3 to 6 independent experiments and expressed as compared to untreated SARS CoV-2-infected Vero-E6 cells. **b** SARS-CoV-2 titer reduction in Vero-E6. Cells were infected with SARS-CoV-2/PSL2020 P#2 stock in triplicate at a multiplicity of infection (MOI) of 0.1 in the presence of different concentrations (from 0.01 to 10 μM) of peptides **P8, P9,** and **P10** for 2 h. Then the virus was removed, and cultures were washed and incubated for 72 h to measure virus production by plaque assay. **c** Cell cytotoxicity in Vero-E6 cells. Cell viability was measured by MTT assays after treatment with vehicle 0, 0.1, 1, or 10 μM of **P8, P9**, or **P10** for 24, 48, or 72 h. Cell death was measured by flow cytometry using annexin-V-APC and PI staining in cells treated with vehicle or 10 μM **P8, P9**, or **P10** for 24, 48, or 72 h. The plots represent the means (±SD) of three independent experiments. **d** SARS-CoV-2 titer reduction in Calu-3. Cells were infected with SARS-CoV-2/PSL2020 P#2 stock in triplicate at a multiplicity of infection (MOI) of 0.3 in the presence of different concentrations (from 0.01 to 10 μM) of peptides **P8, P9**, and **P10** for 2 h, after which the virus was removed, and cultures were washed in, incubated for 72 h to measure virus production by plaque assay. **e** Dose-inhibition curve in Calu-3. Cells were infected with SARS-CoV-2/PSL2020 P#2 stock at respectively a multiplicity of infection (MOI) of 0.3 in the presence of six different concentrations (from 0.01 to 10 μM) of peptides **P1**, **P1 scr**, **P7**, **P8, P9, P10**, for 2 h. Then the virus was removed, and cultures were washed in, incubated for 48 h, before supernatant was collected to measure virus replication by ELISA. Data are combined from 3 to 6 independent experiments and expressed as percent of inhibition compared to untreated SARS CoV-2-infected Vero-E6 cells. Data fitted in the sigmoidal dose-response curve represent the means (±SD) of at least three independent experiments and are expressed as percent of inhibition compared to untreated SARS CoV-2-infected Vero-E6 cells. **f** Cell cytotoxicity in Calu-3 cells. Cell viability was measured by MTT assays after treatment with vehicle 0, 0.1, 1, or 10 μM of **P8, P9**, or **P10** for 24, 48, or 72 h. Cell death was measured by flow cytometry using annexin-V-APC and PI staining in cells treated with vehicle or 10 μM **P8, P9**, or **P10** for 24, 48, or 72 h. The plots represent the means (±SD) of three independent experiments.

## Methods

### General chemistry

*Peptides syntheses*. Peptides were produced manually, synthesized from Fmoc-protected amino acids utilizing standard solid-phase peptide synthesis methods. Solid-phase peptide syntheses were performed in polypropylene Torviq syringes

(10 or 20 mL) fitted with a polyethylene porous disk at the bottom and closed with an appropriate piston. Solvent and soluble reagents were removed through back-and-forth movements. The appropriate protected amino acids were sequentially coupled using PyOxim/Oxyma as coupling reagents. The peptides were cleaved from the rink amide resin with classical cleavage cocktail trifluoroacetic acid/

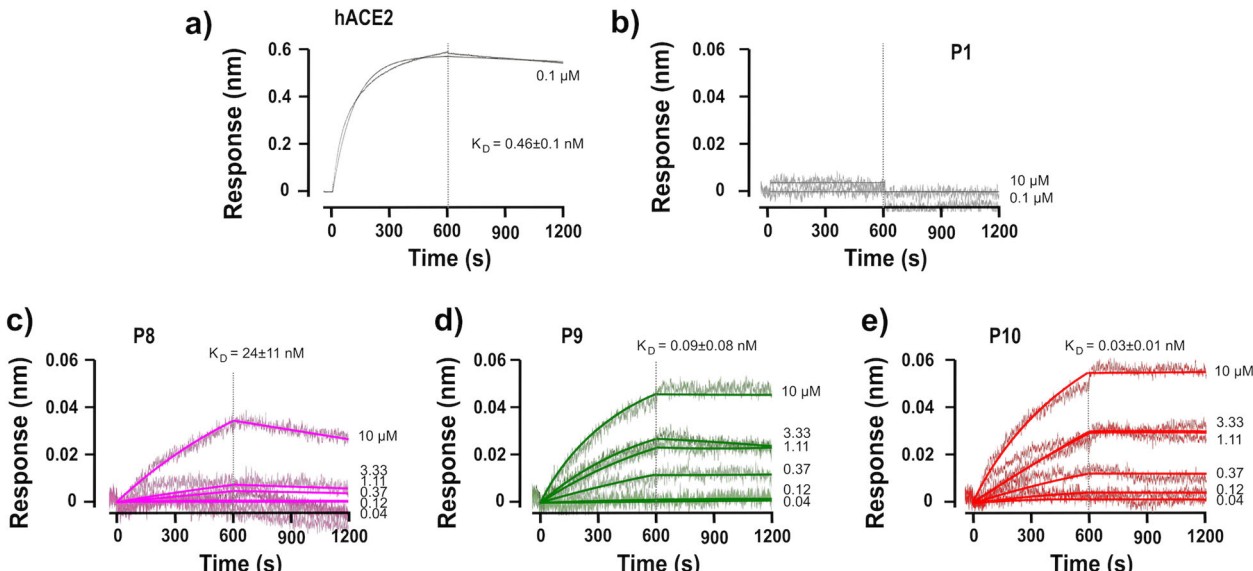

**Fig. 4 Helical peptide-mimics of hACE2 strongly bind to the spike RBD.** The Fc-tagged 2019-nCoV RBD-SD1 (Sanyou Biopharmaceuticals Co. Ltd) was immobilized to an anti-human capture (AHC) sensortip (FortéBio) using an Octet RED96e system (FortéBio). The sensortip was then dipped into **a** 0.1 μM solution of hACE2 (Sanyou Biopharmaceuticals Co. Ltd, His Tag), **b** 0.1 or 10 μM solution of **P1**, or a range of concentrations (1 in 3 dilutions starting from 10 μM, i.e., 41 nM, 123 nM, 370 nM, 1.1 μM, 3.3 μM, and 10 μM) for **c P8**, **d P9**, and **e P10** to measure association rates before being dipped into a well containing only running buffer to measure dissociation rates. Data were reference subtracted and fitted to a 1:1 binding model using Octet Data Analysis Software v10.0 (FortéBio). All figures are representative of at least two independent experiments. $K_D$ values were calculated from Supplementary Table 6.

triisopropyl silane/water (95:2.5:2.5). The crude products were purified using preparative scale high-performance liquid chromatography (HPLC). The final products were characterized by analytical liquid chromatography (LC)–mass spectrometry (MS). All tested compounds were trifluoroacetic acid salts and were at least 95% pure. The relevant peptides after CD spectra analyses were selected and produced by Genecust France on 20 mg scale.

*Purification.* Preparative scale purification of peptides was performed by reverse phase HPLC on a Waters system consisting of a quaternary gradient module (Water 2535) and a dual wavelength UV/visible absorbance detector (Waters 2489), piloted by Empower Pro 3 software using the following columns: preparative Macherey-Nagel column (Nucleodur HTec, C18, 250 mm × 16 mm internal diameter, 5 μm, 110 Å) and preparative Higgins analytical column (Proto 200, C18, 150 mm × 20 mm i.d., 5 μm, 200 Å) at a flow rate of 14 mL min⁻¹ and 20 mL min⁻¹, respectively. Small-scale crudes (<30 mg) were purified using semipreparative Ace column (Ace 5, C18, 250 mm × 10 mm i.d., 5 μm, 300 Å) at a flow rate of 5 mL min⁻¹. Purification gradients were chosen to get a ramp of ~1% solution B per minute in the interest area, and UV detection was done at 220 and 280 nm. Peptide fractions from purification were analyzed by LC−MS (method A or B depending on retention time) or by analytical HPLC on a Dionex system consisting of an automated LC system (Ultimate 3000) equipped with an autosampler, a pump block composed of two ternary gradient pumps, and a dual wavelength detector, piloted by Chromeleon software. All LC−MS or HPLC analyses were performed on C18 columns. The pure fractions were gathered according to their purity and then freeze-dried using an Alpha 2/4 freeze-dryer from Bioblock Scientific to get the expected peptide as a white powder. Final peptide purity (>95%) of the corresponding pooled fractions was checked by LC−MS using method A (outlined below).

*Analytics.* Two methods were conducted for LC−MS analysis. **Method A**. Analytical HPLC was conducted on a X-Select CSH C18 XP column (30 mm × 4.6 mm i. d., 2.5 μm), eluting with 0.1% formic acid in water (solvent A) and 0.1% formic acid in acetonitrile (solvent B), using the following elution gradient: 0−3.2 min, 0−50% B; 3.2 − 4 min, 100% B. Flow rate was 1.8 mL min⁻¹ at 40 °C. The mass spectra were recorded on a Waters ZQ mass spectrometer using electrospray positive ionization [ES + to give (MH) + molecular ions] or electrospray negative ionization [ES−to give (MH) − molecular ions] modes. The cone voltage was 20V. **Method B**. Analytical HPLC was conducted on a X-Select CSH C18 XP column (30 mm × 4.6 mm i.d., 2.5 μm), eluting with 0.1% formic acid in water (solvent A) and 0.1% formic acid in acetonitrile (solvent B), using the following elution gradient: 0−3.2 min, 5−100% B; 3.2−4 min, 100% B. Flow rate was 1.8 mL min⁻¹ at 40 °C. The mass spectra were recorded on a Waters ZQ mass spectrometer using electrospray positive ionization [ES+ to give (MH)+ molecular ions] or electrospray negative ionization [ES− to give (MH)− molecular ions] modes. The cone voltage was 20 V.

**CD spectroscopy**. CD spectra were recorded on a Jasco J-815 CD spectropolarimeter equipped with a Peltier temperature controller. Data were obtained at 25 °C over a wavelength range between 185 nm and 270 nm, using a wavelength interval of 0.2 nm and a scan rate of 20 nm min⁻¹. Peptide samples were prepared at a concentration of 60 μM in 50 mM sodium phosphate buffer, pH 7.4, in a quartz cell of 1 mm path length. CD experiments were processed and plotted with R program. CD spectra were analyzed using DICHROWEB web server and CDSSTR deconvolution algorithm[24].

**Anti-infectivity study on Calu-3 and Vero-E6 cells**

*Cells and virus preparation.* Calu-3 (ATCC HTB55) and Vero-E6 (ATCC CRL-1586) cells were purchased from the American Type Culture Collection and routinely checked for mycoplasma contamination[26]. Cells were cultured in Dulbecco's Modified Eagle Medium (DMEM) supplemented with non-essential amino acids, penicillin-streptomycin, and 10% v/v fetal bovine serum.

The SARS-CoV-2 clinical isolate was obtained from BAL of a symptomatic infected patient (#SARS-CoV-2/PSL2020, available at Pitié-Salpêtrière hospital, Paris (France)). The patient recruited for virus isolation and culture was in intensive care unit in the Pitié Salpêtrière hospital. The patient underwent a bronchoalveolar lavage for clinical purpose (seeking for a bacterial pulmonary infection). The protocol was approved by our institution's ethics committee (Immuno-COVID-REA, CER-Sorbonne Université, no. CER-SU-2020-31). BAL (0.5 mL) was mixed with an equal volume of DMEM without FBS, supplemented with 25 mM Hepes, double concentration of penicillin–streptomycin and miconazole (Sigma), and added to 80% confluent Vero-E6 cells monolayer seeded into a 25-cm² tissue culture flask. After 1 h adsorption at 37 °C, 3 mL of infectious media (DMEM supplemented with 2% FBS, penicillin-streptomycin and miconazole) were added. Twenty-four hours post-infection another 2 mL of infectious media were added. Five days post-infection, supernatants were collected, aliquoted, and stored at −80 °C (P#1). For secondary virus stock, Vero-E6 cells seeded into 25 cm² tissue culture flasks were infected with 0.5 mL of P#1 stored aliquot, and cell-culture supernatant were collected 48 h post-infection and stored at −80 °C (P#2). Infectious viral particles were measured by a standard plaque assay previously described[25] with fixation of cells 72 h post infection. Accordingly, the viral titer of SARS-CoV-2/PSL2020 P#2 stock was about 5.3 10⁵ PFU mL⁻¹.

*Viral neutralization.* Vero-E6 or Calu-3 (1 × 10⁵ cells mL⁻¹) were seeded into 24 wells plates in infectious media and treated with different concentrations of the peptides (from 0.1 to 10 μM). After 30 min at room temperature, cells were infected with 0.1 multiplicity of infection (MOI) (Vero-E6) or 0.3 MOI (Calu-3) of SRAS-CoV-2 (SARS-CoV-2/PSL2020 P#2 stock) in infectious media. Cell supernatants were collected at 48 h post-infection for enzyme-linked immunosorbent assay (ELISA) assay using a SARS-CoV-2 (2019-nCoV) nucleoprotein ELISA kit (Sino biological), according the manufacturer's instructions, and standard plaque assay[26].

Inhibition of infection was calculated comparing viral concentration in each case with that of untreated SARS- CoV-2-infected cells.

**Toxicity study on Calu-3 and Vero-E6 cells**. Cell cytotoxicity was measured using MTT assays and fluorescent activated cell sorting (FACS). Calu-3 and Vero-E6 were seeded in 96-well (for MTT assays) or 24-well (for FACS) plates and let adhere. When ~40% confluence was reached, the wells were washed, and new medium containing vehicle (water, 5% final volume) or different concentrations of the indicated peptide was added. MTT (2 mM) was added to each well after 24 h, 48 h, or 72 h of treatment and incubated 4 h at 37 °C. Supernatants were then discarded and formazan salts were dissolved in DMSO to read plate absorbance at 570 nM. Absorbance in each well was normalized with those treated with vehicle. For FACS analyses, cells were harvested from the wells with the help of accutase, pelleted, and stained with Annexin-V-APC (0.1 µg mL$^{-1}$) and propidium iodide (0.5 µg mL$^{-1}$) in annexin binding buffer (10 mM Hepes pH 7.4, 140 mM NaCl, and 2.5 mM CaCl$_2$). Cells were sorted in a FACScalibur flow cytometer and data was analyzed using FlowJo 10.0 software, considering cell death as the sum of Ann-V$^+$/PI$^-$ and Ann-V$^+$/PI$^+$ events.

**Biolayer Interferometry experiments**. Fc-tagged 2019-nCoV RBD-SD1 (Sanyou Biopharmaceuticals Co. Ltd) was immobilized to an anti-human capture sensortip (FortéBio) using an Octet RED96e (FortéBio). The sensortip was then dipped into 100 nM hACE2 (Sanyou Biopharmaceuticals Co. Ltd, His Tag) or 1 µM of any tested peptide to measure association before being dipped into a well containing only running buffer composed of DPBS (Potassium Chloride 2.6 mM, Potassium Phosphate monobasic 1.5 mM, Sodium Chloride 138 mM, Sodium Phosphate dibasic 8 mM), 0.05% Tween 20 and 0.5% bovine serum albumin to measure dissociation. (Extended conditions, and suppliers in Supplementary Note 1).

**Statistics and reproducibility**. All conditions in the viral neutralization and cell cytotoxicity experiments were tested in triplicate. Data from viral neutralization assays was analyzed using nonlinear regression, fitting sigmoidal curves used to calculate the IC$_{50}$ values with Graphpad Prism 8.0.1. For BLI experiments, data were reference subtracted and fit to a 1:1 binding model using Octet Data Analysis Software v10.0 (FortéBio). $K_D$ values were extracted from 3 independent determinations (**P9** and **P10**) validated by a statistical coefficient determination ($R^2$) > 0.95.

**Reporting summary**. Further information on research design is available in the Nature Research Reporting Summary linked to this article.

## Data availability
The complex between hACE2 and the surface spike protein of SARS-CoV-2 data used for peptide design is available in RCSB PDB with the identifier doi: 10.2210/pdb6M0J/pdb[14]. The randomly selected RBD amino-acid sequences from China (QOH25833, QIG55857, QHR63290, QHR63250, QJG65957, QJG65956, and QJG65951), USA (QIV65044, QJD23847, QJU11481, QJD24531, QJD25193, QJD25529, and QJA17180), and France (QJT73034, QJT73010, QJT72902, QJT72806, QJT72794, QJT72722, QJT72710, QJT72626, QJT72614, and QJT72554), used for sequence alignment are available in the Genebank database at ncbi.nlm.nih.gov. The amino acid sequence of the Fc-tagged 2019-nCoV RBD-SD1 was provided by Sanyou bio with purchase (sanyoubio.com/EN/2019-nCoV.php) and is depicted in Supplementary Fig. 1a. The datasets generated and/or analyzed during the current study are in the manuscript and the raw data are available from the corresponding author on reasonable and justified request.

## Code availability
The predictive data supporting our work was generated using publicly available software. For sequence alignment, the Weblogo software (available at https://weblogo.berkeley.edu) and Clustal Omega (available at https://ebi.ac.uk/Tools/msa/clustalo/) were used. The Agadir helical content predictor is available at https://agadir.crg.es. Peptide antigenicity can be determined at https://imed.med.ucm.es/Tools/antigenic.pl. Online analysis for peptide CD spectra and using DichroWeb server and the CDSSTR algorithm is available at https://dichroweb.cryst.bbk.ac.uk. The HELIQUEST web server used to calculate peptide hydrophobicity and hydrophobic moment is available at https://heliquest.ipmc.cnrs.fr.

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

## Acknowledgements
This work was supported by private funds (P.K.), SATT-Lutech, Kaybiotix (LGM PhD grant), and French Research Ministry (EO PhD grant). P.K. is grateful to SATT-Lutech team supporting this project, to Fabrice Viviani and Akanksha Gangar from Oncodesign for their unwavering support. The authors thank David Boutolleau from the Virology Department, Pitié-Salpêtrière, AP-HP, where was diagnosed the BAL SARS-COV-2 infection. Philippe Karoyan dedicates this work to Gérard Chassaing on the occasion of his 75th birthday. "There are no borders in science. The only limit is our imagination."

## Author contributions
P.K. conceived and supervised this project, designed and synthesized the peptides, designed the experiments, interpreted the data, and wrote the draft and the discussion of the manuscript. P.K. and O.L. wrote the manuscript. P.K. and O.L. performed the molecular modeling study. A.D. contributed to the molecular modeling study. O.L. performed the CD structural studies. V.V. performed the cell inhibition assays. P.G. performed the binding experiments. E.O. performed the peptides LC-MS analyses. L.G.M. performed the cell toxicity experiments. A.G. and C.E.L. recruited the patient and provided BAL sample

## Competing interests
The authors declare the following competing financial interest(s): The patent application EP20305449.9 included results from this paper.
