## [Peer Review File · Communications Biology]

Reviewers' comments:

Reviewer #1 (Remarks to the Author):

In their manuscript entitled, "Human ACE2 peptide mimics block SARS-CoV-2 Pulmonary Cell infection," Karoyan et al., present a unique and exciting method for inhibiting SARS-CoV-2 by means of a peptide mimetic approach (using all-natural amino acids). Cleverly, the authors inspected the spike protein-ACE2 receptor atomic structure and devised a method wherein peptide mimics would be created from H1 of ACE2 to bind to and block a portion of the spike protein required for binding and infectivity. The authors created over a dozen peptide mimics in the 10-15 amino acid range that were optimized not only for low energy intermolecular interactions but also for alpha helicity (using nice bioinformatics) and solubility. The peptide P10 appears to inhibit viral replication by 100% at 10uM. This was matched by using an in vitro binding assay known as BLI (not very different from SPR) to gain KD's of the purified peptides which ranged in the low nanomolar to picomolar range. The peptide synthesis and purification are of high quality along with the CD. The viral replication assays appear to be of equally high quality but this is outside my area of expertise. This is a potentially very important discovery that could lead to later therapeutics and I think it should be published with just minor corrections listed below.

English language is generally quite good but some sections had poor usage of articles and other issues. Please go through one more time.

Figure 1 is really noisy and there are not good labels. Maybe some of the side chain sticks can be hidden or zoom-ins of critical interactions could be shown in a panel.

Similarly, the interactions weren't described in sufficient detail. For example, page 3, "This alphahelix composed of 27 residues contains 12 residues involved in hydrophobic interactions, hydrogen bonds, and salt bridges." Such as?

Was the sequence known for the clinical virus you obtained? Was this modelled in the interaction poses?

Several questions regarding the BLI. It looks like kinetics were derived from one plot (4a), ie Kd and Ka. It seems like one plot is not a robust way to do this. Shouldn't there be multiple concentrations?

hACE2 was dipped at 100nM but the KDs for the peptides were much tighter. Shouldn't the KD be near the hACE2 concentration with dose points on left and right of that number. Was this at least repeated more than once? There is quite a bit of drift on P1 sensorgram. Why is this?

In the discussion it is stated that this pandemic drug cannot be a small molecule. I tend to agree; however, that's a fairly big claim and should be backed up.

Why were only natural amino acids used? Would non-natural amino acids be better?

What about susceptibility to proteases? For the peptide mimics wouldn't it be good to do plasma half life experiments?

Reviewer #2 (Remarks to the Author):

This manuscript presents the design and evaluation of peptides derived from the N-terminal helix of ACE2, which is the primary cellular receptor for the entry of SARS-CoV-2. The N-terminal ACE2 helix

contains most of the contact residue of ACE2 for its interaction with the spike protein on the surface of SARS-CoV-2, through which the virus attaches itself to the cell. The peptides were designed and optimized using computer algorithms aimed at improving helicity, as well as minimizing immunogenicity of peptides. The resulting peptides appear to be able to bind with high affinity to the receptor binding domain (RBD) of the spike protein, as well as to inhibit infection of cells with SARS-CoV-2. This is an outstanding result, considering that a peptide presenting the wt ACE2 helix is neither helical, nor does it bind to spike RBD or inhibit SARS-CoV-2 infection. The designed peptides, in particular P10, could now serve as starting point for novel therapeutic approaches to treat COVID-19 patients, possibly by topical application of the peptide, which appears to be non-toxic to pulmonary cells, through inhalation.

I do, however, have substantial concerns regarding the execution and presentation of the experiments and their results, which I believe are not in accordance with current scientific standards.

According to Figs. 3C and 3D, inhibition of SARS-CoV-2 replication in Vero and Calu3 cells, respectively, was tested at only four concentrations (10, 1, 0.1 and 0.01 μM). Such a limited data set within that large a range of concentration (three logs) is insufficient to correctly calculate IC₅₀ values. For that, the peptides would need to be tested at more concentrations within that range, resulting in a sufficient number of data points, which should then be fitted to a sigmoidal dose-response-curve using non-linear regression analysis, enabling correct calculation of IC₅₀ values.

Furthermore, it is mathematically incorrect to calculate K_D values based on only one tested concentration. More specifically, if that single concentration is 1 μM (Fig. 4B), calculated K_D values in the range of 20 pM simply cannot be correct. Moreover, K_Ds for P9 and P10 are given as 21 ± 191 pM and 25 ± 117 pM, respectively (Fig. 4B), so the deviation is five to ten-fold higher than the average value - from how many replicates were these values calculated? In Table 1, on the other hand, the K_Ds of the same peptides are given simply as < 1 nM.

Based on the above points, I can't help but wonder about the accuracy of the data presented in this manuscript.

Overall, the results reported in this manuscript may potentially be of high value to the scientific community, as well as for the development of efficient therapies against the COVID-19 pandemic. The manuscript does, however, require extensive revision, as indicated above, which includes essential additional experiments, in order to meet the high standard of Communications Biology.

Philippe KAROYAN, Professeur

Nature Communications Biology

Paris, 2020, December 1st,

Point by point responses to reviewers' comments

All modifications, corrections and new data have been highlighted in Yellow in the main text of the manuscript.

Some answers addressed in the initial manuscript were highlighted in green in the main text of the manuscript.

A-Answers (A1-9) to R#1 questions (Q1-9):

Q1-English language is generally quite good but some sections had poor usage of articles and other issues. Please go through one more time.

A1: As requested by R#1, the manuscript was carefully revised for its English language and corrected by a colleague from the United States.

Q2-Figure 1 is really noisy and there are not good labels. Maybe some of the side chain sticks can be hidden or zoom-ins of critical interactions could be shown in a panel.

A2: As requested by R#1, some of the side chains not directly involved in the interaction have been hidden, the side chain labelling has been improved and the figure 1B has been zoomed to highlight the critical interactions as followed in the manuscript:

Figure 1. Structure of the complex between hACE2 and the spike protein of SARS-CoV-2 (pdb 6m0j).¹⁴ a) Contact residues of the hACE2 / SARS-CoV-2 spike interface. hACE2 protein is colored in

Philippe KAROYAN, Professeur

green, apart from N-terminal helix H1 which is highlighted in salmon. SARS-CoV-2 spike protein is shown in cyan. **b)** Residues of hACE2 H1 helix interacting with spike. **c)** Sequence of hACE2 H1 helix showing the 12 interacting residues in magenta. Residue positions in green were considered as possible substitution sites for the helical peptide design.

Q3-Similarly, the interactions weren't described in sufficient detail. For example, page 3, "This alphahelix composed of 27 residues contains 12 residues involved in hydrophobic interactions, hydrogen bonds, and salt bridges." Such as?

A3: to answer this question, a **supplementary Table 1** has been added in the supporting information describing all the contact residues involved un hACE2/SARS-CoV-2 Spike interaction.

Supplementary Table 1

Supplementary table 1. Residues involved in ACE2 / SARS-CoV2 Spike interaction				
ACE2	Residue	Interactions with SARS-CoV-2 Spike residues		
		Hydrogen bond	Salt bridge	Van der Waals
H1	Q24	N487 (OE1-ND2 2.7 Å)		
	T27			F456, A475, Y489
	F28			Y489
	D30	K417 (OD2-NZ 2.9 Å)	K417 (OD2/OD1-NZ 2.9/4.0 Å)	
	K31	Q493 (NZ-NE2 2.9 Å)	E484 (NZ-OE1 4.4 Å)	F456, Y489
	H34			Y453, L455
	E35	Q493 (OE1/OE2-NE2 3.1/3.5 Å)		
	E37	Y505 (OE2-OH 3.5 Å)		
	D38	Y449 (OD2/OD1-OH 2.7/3.2 Å)		
	Y41	T500 (OH-OG1 2.7 Å) N501 (OH-N 3.7 Å)		Q498
	Q42	G446 (NE2-O 3.2 Å) Y449 (NE2-OH 2.8 Å) Q498 (OE1-NE2 3.4 Å)		
	L45			Q498, T500
H2	L79			F486
	M82			F486
	Y83	N487 (OH-OD1 2.8 Å) Y489 (OH-OH 3.5 Å)		F486
	N330			T500
	K353	G496 (NZ-O 3.1 Å) G502 (O-N 2.8 Å)		Y505
	G354			G502
	D355	T500 (OD2-O 3.3 Å)		
	R357	T500 (NH1-OG1 3.7 Å)		
	R393	Y505 (NH2-OH 3.7 Å)		

H1 and H2, Helix 1 and 2; ND2, nitrogen delta 2; NE2, nitrogen epsilon 2; NZ, nitrogen zeta; N, nitrogen; NH1, nitrogen eta 1; NH2, nitrogen eta 2; OH, oxygen eta; O, oxygen; OD1, oxygen delta 1; OD2,

Philippe KAROYAN, Professeur

oxygen delta 2; OG1, oxygen gamma 1; OE1, oxygen epsilon 1; OE2, oxygen epsilon 2.
--

This information was added as follows in the main text of the manuscript:

This α -helix (**Figure 1b**), composed of 27 residues (from S19 to L45, **Figure 1c**) contains 12 residues (highlighted in magenta in **Figure 1c**) involved in hydrogen bonds, salt bridges and van der Waals interactions (see **Supplementary Table 1** for details).

Q4- Was the sequence known for the clinical virus you obtained? Was this modelled in the interaction poses?

A4: the sequence of the clinical virus strain isolated in France, Pitié-Salpêtrière Hospital (SARS-CoV-2/PSL2020 P2) was not known and not used for the modelling at this stage. Whatever, we performed a Clustal Multiple Sequence alignment of viruses isolated in China (7 sequences analyzed), United States (7 sequences analyzed) and France (10 sequences analyzed) and compared these sequences with the SARS-CoV-2 RBD sequence protein bought from Sanyou Bio (<https://www.sanyoubio.com/EN/2019-nCoV.php>). From these analyses, we observed that all the Spike proteins sequences of the SARS-CoV-2 viruses from China, US and France were 100% identical to the sequence used for our structural studies, at least in the interacting interfaces. This highlights a highly conserved sequence for the portion of Spike interacting with the α -helix-H1 of hACE2, possibly because deleterious mutations for the virus in this interface would prevent its infectious nature.

The following supplementary **figure S1** has been added to the supporting information file:

Supplementary Figure 1

Supplementary Figure 1

Philippe KAROYAN, Professeur

Supplementary Figure 1. Highly conserved sequence of the SARS-CoV-2 RBD motif. a) Complete sequence of the SARS-CoV-2-huFc fusion protein used in the present study (Sanyou Bio, reference PN003). The RBD motif is highlighted in bold, the rest represent the huFc fragment. **b)** Sequence alignment of 24 random RBD amino acid sequences randomly listed on NCBI resources from China (QOH25833, QIG55857, QHR63290, QHR63250, QJG65957, QJG65956, QJG65951), USA (QIV65044, QJD23847, QJU11481, QJD24531, QJD25193, QJD25529, QJA17180), and France (QJT73034, QJT73010, QJT72902, QJT72806, QJT72794, QJT72722, QJT72710, QJT72626, QJT72614, QJT72554), and the RBD motif from the SARS-CoV-2-huFc fusion protein. The alignment was performed using the Weblogo software (<http://weblogo.berkeley.edu/>) and validated using Clustal Omega software. Both confirmed one point-mutation (V53E) in one (QJT72806) out of the 24 sequences. **c)** Structure of the complex between hACE2 and the Spike protein of SARS-CoV-2 (pdb 6m0j) highlighting the V53 side chain in red sphere.

The following comment has been added in the main text of our manuscript:

A Clustal multiple sequence alignment of viruses isolated in China, United States and France was performed (See **Supplementary Figure 1**). From these analyses, we observed that all the randomly selected sequences were 100% identical at least in the ACE2 interacting interface. This highlights a highly conserved sequence for the portion

Philippe KAROYAN, Professeur

of spike interacting with the α -helix H1 of hACE2, possibly because deleterious mutations at this interface would limit viral infectivity.

Q5-Several questions regarding the BLI. It looks like kinetics were derived from one plot (4a), ie K_D and K_a . It seems like one plot is not a robust way to do this. Shouldn't there be multiple concentrations? hACE2 was dipped at 100nM but the K_D s for the peptides were much tighter. Shouldn't the K_D be near the hACE2 concentration with dose points on left and right of that number. Was this at least repeated more than once?

A5: The representative plots in our initial manuscript were realized based on 2 and 3 independent experiments but this information was missing in the manuscript and we thank R#1 for highlighting this point.

Whatever, to answer this important question also highlighted by R#2, we performed multiple concentrations experiments (41 nM, 123 nM, 370 nM, 1.1 μ M, 3.3 μ M and 10 μ M), displaying a concentration-dependent signal increase as reported on the following figures. All figures are representative of at least two independent experiments and reported in now five plots with 6 different concentrations each:

Figure 4. Helical peptide-mimics of hACE2 strongly bind to the spike RBD. The Fc-tagged 2019-nCoV RBD-SD1 (Sanyou Biopharmaceuticals Co. Ltd) was immobilized to an anti-human capture (AHC) sensortip (FortéBio) using an Octet RED96e system (FortéBio). The sensortip was then dipped into **a)** 0.1 μ M solution of hACE2 (Sanyou Biopharmaceuticals Co. Ltd, His Tag), **b)** 0.1 or 10 μ M solution of **P1**, or a range of concentrations (1 in 3 dilutions starting from 10 μ M, i.e., 41 nM, 123 nM, 370 nM, 1.1 μ M, 3.3 μ M and 10 μ M) for **c)** **P8**, **d)** **P9** and **e)** **P10** to measure association rates before being dipped into a well containing only running buffer to measure dissociation rates. Data were reference subtracted and fitted to

Philippe KAROYAN, Professeur

a 1:1 binding model using Octet Data Analysis Software v10.0 (FortéBio). All figures are representative of at least two independent experiments. K_D values were calculated from **Supplementary Table 6**.

The following modifications were done in the main text:

The designed peptides bind to SARS-CoV-2 spike RBD with high affinity

Finally, the peptides that were able to block cell infection with an IC_{50} in the sub- μ M range (**P8**, **P9** and **P10**) were evaluated for their ability to bind to SARS-CoV-2 spike RBD (**Figure 4**) using biolayer Interferometry (BLI) with an Octet RED96e system (FortéBio).³⁰ hACE2 was used as a positive control (**Figure 4a**).

Even though this technique presents some drawbacks³¹ offering narrow signal windows with low molecular weight analytes³² such as peptides, it remained useful to identify and rank our binding mimics. In the conditions tested here, peptide **P1** does not bind to RBD when using 100 nM nor 10 μ M peptide solutions (**Figure 4b**). For peptides **P8**, **P9** and **P10**, multiple concentrations experiments were performed and dose-dependent associations were observed (**Figure 4c-e**). Of note, only association rates could be quantified accurately, the dissociation ones being very slow, highlighting strong binding properties for these mimics (**supplementary Table 6**).

Q6-There is quite a bit of drift on P1 sensorgram. Why is this?

A6- When dealing with very low signal, a drift can be observed in some experiments even with reference sample (buffer) and reference sensor (unloaded with RBD) correction process, as this was the effective case for the non-active peptide **P1**.

Q7-In the discussion it is stated that this pandemic drug cannot be a small molecule. I tend to agree; however, that's a fairly big claim and should be backed up.

A7-We do agree and the statement “can’t be” has been modified to “... developing a specific drug at a pandemic speed is a hard task especially in the design of a small molecule. Indeed, beyond the time required for the identification and validation of a lead compound after a library screening, followed by structure-activity relationship studies and clinical development, small molecule drugs are associated with a high attrition rate partly due to their off-target toxicity observed during pharmacological studies.”

Q8-Why were only natural amino acids used? Would non-natural amino acids be better?

Philippe KAROYAN, Professeur

A8-We thank R#1 for this fundamental question. We are aware that using only natural amino acids to stabilize a helical structure was already a challenge by itself. Stabilizing secondary structures with constrained amino acids is well documented and validated, using for example stapled peptides strategy. However, these approaches require long and tedious Structure-activity-relationship studies not compatible with the design of a drug at a pandemic speed. As explained in our manuscript, our choice was guided by the desire to build a simple peptide easy to produce quickly on a large scale, without technical constraints requiring sometimes laborious development. We thought it would be a fair compromise between designing α -helix peptides with optimized binding affinity and developing an effective tool within short deadlines while integrating the constraints of developability to move a prophylactic device and/or a therapeutic peptide drug quickly in the clinic.

Moreover, as specialists of non-natural amino acid design and development, we are aware of the costs of goods associated with the production of such tools on an industrial scale, costs which could appear prohibitive in the context of industrial development.

Q9-What about susceptibility to proteases? For the peptide mimics wouldn't it be good to do plasma half-life experiments?

A9- We thank R#1 for highlighting this crucial point. Indeed, intrinsic limitations for therapeutic peptides include metabolic instability, that is, the inability to withstand 600 proteases in the human body and restriction to the parenteral route of administration. Chemically synthesized peptides can overcome these challenges by incorporating additional entities, such as non-natural amino acids, to improve the metabolic stability, or other chemical entities, such as polyethylene glycols, to enhance membrane transportation. Nevertheless, as explained in our manuscript (see conclusion) our aim is to formulate the designed peptide as a medical device of class 1 designed as a nasal spray for prophylaxis. The peptide will not enter the blood vessels: plasma half-life experiments might be realized but seem unnecessary for our purpose. In fact, in our case, a reasonable instability is an asset to develop a medical device of class 1. Increasing the plasma stability would lead to classification of our peptide as a “drug” no longer allowing its use as part of a medical device.

All these points are highlighted in the conclusion of the discussion of the manuscript:

Targeting prophylaxis first might shorten the drug development time scale. Delivered through a medical device such as a nasal or oral spray or as a sublingual tablet, these peptides could be aimed at blocking the infectivity of the virus in a preventive manner. Their biodistribution would be limited to the upper airways (nasal and oral cavity) before they are degraded in the digestive tracks without toxic residues. Their therapeutic use might also be considered formulated in that later case as an inhaler to reach pulmonary cells.

Philippe KAROYAN, Professeur

Reviewer #2 (Remarks to the Author):**B-Answers (A1-2) to R#2 questions (Q1-2):**

Q1-According to Figs. 3C and 3D, inhibition of SARS-CoV-2 replication in Vero and Calu3 cells, respectively, was tested at only four concentrations (10, 1, 0.1 and 0.01 μM). Such a limited data set within that large a range of concentration (three logs) is insufficient to correctly calculate IC₅₀ values. For that, the peptides would need to be tested at more concentrations within that range, resulting in a sufficient number of data points, which should then be fitted to a sigmoidal dose-response-curve using non-linear regression analysis, enabling correct calculation of IC₅₀ values.

Answer 1: We do agree and thank R#2 for this remark. In order to improve the quality of our results and in agreement with our project, the inhibition experiments were repeated for Calu3 cell line with ten different concentrations instead of four, i.e. 0.01 μM , 0.03 μM , 0.06 μM , 0.1 μM , 0.3 μM , 0.6 μM , 1.0 μM , 3.3 μM , 6.6 μM and 10 μM , resulting in a sufficient number of data points, which were then fitted to a sigmoidal dose-response-curve using non-linear regression analysis, enabling us to calculate more precise IC₅₀ values and reported in a new figure 3 :

Philippe KAROYAN, Professeur

Figure 3. Peptide mimics of hACE2 show high anti-infective efficacy and are devoid of cell toxicity. **a.** Percent inhibition of SARS-CoV-2 replication. Vero-E6 cells were infected with SARS-CoV-2/PSL2020 P#2 stock at a multiplicity of infection (MOI) of 0.1 in the presence of 10 peptides (**P1 to P8**, **P1scr** and **Ppen** as controls) at 10 μM for 2 h. Then, the virus was removed, and cultures were washed, incubated for 48 h, before supernatant was collected to measure virus replication by ELISA. Data are combined from 3 to 6 independent experiments and expressed as compared to untreated SARS CoV-2-infected Vero-E6 cells. **b.** SARS-CoV-2 titer reduction in Vero-E6. Cells were infected with SARS-CoV-2/PSL2020 P#2 stock in triplicate at a multiplicity of infection (MOI) of 0.1 in the presence of different concentrations (from 0.01 to 10 μM) of peptides **P8**, **P9** and **P10** for 2 h. Then the virus was removed, and cultures were washed and incubated for 72 h to measure virus production by plaque assay. **c.** Cell cytotoxicity in Vero-E6 cells. Cell viability was measured by MTT assays after treatment with vehicle 0, 0.1, 1 or 10 μM of **P8**, **P9** or **P10** for 24, 48, or 72 h. Cell death was measured by flow cytometry using annexin-V-APC and PI staining in cells treated with vehicle or 10 μM **P8**, **P9** or **P10** for 24, 48, or 72 h. The plots represent the means (±SD) of three independent experiments. **d.** SARS-CoV-2 titer reduction in Calu-3. Cells were infected with SARS-CoV-2/PSL2020 P#2 stock in triplicate at a multiplicity of infection (MOI) of 0.3 in the presence of different concentrations (from 0.01 to 10 μM) of peptides **P8**, **P9** and **P10** for 2 hr, after which the virus was removed, and cultures were washed in, incubated for 72 h to measure virus production by plaque assay. **e.** Dose-inhibition curve in Calu-3. Cells were infected with SARS-CoV-2/PSL2020 P#2 stock at respectively a multiplicity of infection (MOI) of 0.3 in the presence of 6 different concentrations (from 0.01 to 10 μM) of peptides **P1**, **P1 scr**, **P7**, **P8**, **P9**, **P10**, for 2 h. Then the virus was

Philippe KAROYAN, Professeur

removed, and cultures were washed in, incubated for 48 h, before supernatant was collected to measure virus replication by ELISA. Data are combined from 3 to 6 independent experiments and expressed as percent of inhibition compared to untreated SARS CoV-2-infected Vero-E6 cells. Data fitted in the sigmoidal dose-response curve represent the means (\pm SD) of at least three independent experiments and are expressed as percent of inhibition compared to untreated SARS CoV-2-infected Vero-E6 cells. f. Cell cytotoxicity in Calu-3 cells. Cell viability was measured by MTT assays after treatment with vehicle 0, 0.1, 1 or 10 μ M of **P8**, **P9** or **P10** for 24, 48, or 72 h. Cell death was measured by flow cytometry using annexin-V-APC and PI staining in cells treated with vehicle or 10 μ M P8, P9 or P10 for 24, 48, or 72 h. The plots represent the means (\pm SD) of three independent experiments.

Q2-Furthermore, it is mathematically incorrect to calculate KD values based on only one tested concentration. More specifically, if that single concentration is 1 μ M (Fig. 4B), calculated KD values in the range of 20 pM simply cannot be correct. Moreover, KDs for P9 and P10 are given as 21 ± 191 pM and , 25 ± 117 pM, respectively (Fig. 4B), so the deviation is five to ten-fold higher than the average value - from how many replicates were these values calculated? In Table 1, on the other hand, the KDs of the same peptides are given simply as < 1 nM. Based on the above points, I can't help but wonder about the accuracy of the data presented in this manuscript.

A2- We of course agree with this remark of R#2. As very recently written by Cao & al. in Science 370, 426–431 (2020), “dissociation constants (Kd) could not be accurately estimated using BLI because of a lack of instrument sensitivity and long equilibration time” below 200 pM. The lack of instrument sensitivity is accentuated in the case of low molecular weight analyte as one's can observed in the work reported by Pinto, et al. in Nature, 583, 290-294 (2020), (Cross-neutralization of SARS-CoV-2 by a human monoclonal SARS-CoV antibody). Using BLI to evaluate affinity, Pinto & al. reported Kd without any standard deviation, using increasing amount of sample relatively to their molecular weight, exemplifying the difficulty of using smaller molecules to measure affinity.

In our case, the BLI was used as a qualitative method to qualify the peptides of interest, identified from their ability to block viral infection on cell-based assays. The observed association and dissociation rates were used for observed affinity calculation in a standard procedure which was 1 μ M peptide dipping. Such a procedure was used comfortably to qualify and rank peptides for further correlations (e.g. with cell-based assay potencies).

Anyway, as requested, we performed multiple concentrations experiments (41 nM, 123 nM, 370 nM, 1.1 μ M, 3.3 μ M and 10 μ M), reported on the following figures. All figures are representative of at least two independent experiments and reported in now five plots with 6 different concentrations for peptides **P8**, **P9** and **P10**:

Philippe KAROYAN, Professeur

Figure 4. Helical peptide-mimics of hACE2 strongly bind to the spike RBD. The Fc-tagged 2019-nCoV RBD-SD1 (Sanyou Biopharmaceuticals Co. Ltd) was immobilized to an anti-human capture (AHC) sensortip (FortéBio) using an Octet RED96e system (FortéBio). The sensortip was then dipped into **a)** 0.1 μM solution of hACE2 (Sanyou Biopharmaceuticals Co. Ltd, His Tag), **b)** 0.1 or 10 μM solution of **P1**, or a range of concentrations (1 in 3 dilutions starting from 10 μM , i.e., 41 nM, 123 nM, 370 nM, 1.1 μM , 3.3 μM and 10 μM) for **c)** **P8**, **d)** **P9** and **e)** **P10** to measure association rates before being dipped into a well containing only running buffer to measure dissociation rates. Data were reference subtracted and fitted to a 1:1 binding model using Octet Data Analysis Software v10.0 (FortéBio). All figures are representative of at least two independent experiments. K_D values were calculated from **Supplementary Table 6**.

K_D values obtained here are reported in supplementary Table 6:
Supplementary Table 6

	Sample ID	Conc. (nM)	Response	K_D (M)	K_D Error	k_a (M s^{-1})	k_a Error	k_{dis} (s^{-1})	k_{dis} Error	Full R^2
EXP1	ACE2	10	0.3118	3.94E-10	5.63E-12	15.78E+04	4.79E+02	6.22E-05	8.68E-07	0.9989
EXP2	ACE2	100	1.0175	4.13E-10	2.66E-11	8.00E+04	2.50E+02	3.30E-05	2.12E-06	0.9801
EXP3	ACE2	100	0.6917	5.72E-10	2.33E-11	8.97E+04	2.77E+02	5.13E-05	2.08E-06	0.9783
EXP1	P8	100	0.0218	1.59E-08	1.26E-09	5.61E+03	5.95E+03	8.94E-05	6.99E-06	0.8764
EXP2	P8	1000	0.0191	3.17E-08	2.02E-09	4.31E+03	6.10E+01	1.37E-04	8.47E-06	0.8687
EXP1	P9	1000	0.0209	4.74E-11	5.40E-10	1.02E+03	3.31E+01	<1.0E-07	5.50E-07	0.9304
EXP2	P9	1111	0.0203	4.38E-11	4.50E-10	1.10E+03	2.71E+01	<1.0E-07	4.95E-07	0.9381
EXP3	P9	10000	0.0406	1.90E-10	1.48E-09	2.54E+02	2.50E+00	<1.0E-07	3.76E-07	0.9608
EXP1	P10	1000	0.046	1.94E-11	1.39E-10	2.49E+03	2.28E+01	<1.0E-07	3.45E-07	0.9697

Philippe KAROYAN, Professeur

EXP2	P10	1111	0.0282	3.43E-11	4.19E-10	1.41E+03	3.29E+01	<1.0E-07	5.91E-07	0.9248
EXP3	P10	1000	0.0446	3.36E-11	3.70E-10	1.43E+03	3.27E+01	<1.0E-07	5.31E-07	0.9422

K_D , equilibrium dissociation constant; k_a , association constant; k_{dis} , dissociation constant; R^2 , coefficient of determination.

From these experiments, we observed that peptides **P8**, **P9** and **P10** are displaying a concentration-dependent signal increase, highlighting a good correlation between peptides structure, activities and binding properties. Indeed, peptide without defined structure such as **P1** does not show any binding neither any ability to block SARS-CoV-2 viral infection.

Even though this technique presents some drawbacks offering narrow signal windows with low molecular weight analytes such as peptides compared to hACE2 protein, the binding characteristics of our positive peptides are confirmed.

A quite slow association rate is observed compared to the one observed with hACE2 but dissociation rates are confirmed to be very low: these observations can be interpreted as being the result of a slow association leading to a strong interaction preventing dissociation under our experimental conditions, highlighting the peculiar behavior of our active peptides, literally sticking to spike RBD.

REVIEWERS' COMMENTS:

Reviewer #1 (Remarks to the Author):

I believe questions have been answered adequately and can be now published.

Reviewer #2 (Remarks to the Author):

I appreciate the modifications and additions in the revised manuscript, which were done in response to my first review, and I now recommend publication of this manuscript in Communications Biology